# Kidney Damage Caused by Obesity and Its Feasible Treatment Drugs

**DOI:** 10.3390/ijms23020747

**Published:** 2022-01-11

**Authors:** Meihui Wang, Zixu Wang, Yaoxing Chen, Yulan Dong

**Affiliations:** 1Neurobiology Laboratory, Department of Basic Veterinary Medicine, College of Veterinary Medicine, China Agricultural University, Beijing 100193, China; wangmeihui718@163.com (M.W.); zxwang2007@163.com (Z.W.); yxchen@cau.edu.cn (Y.C.); 2Key Laboratory of Precision Nutrition and Food Quality, Ministry of Education, Department of Nutrition and Health, China Agricultural University, Beijing 100193, China

**Keywords:** obesity, kidney damage, treatment, melatonin, inflammation, oxidative stress

## Abstract

The rapid growth of obesity worldwide has made it a major health problem, while the dramatic increase in the prevalence of obesity has had a significant impact on the magnitude of chronic kidney disease (CKD), especially in developing countries. A vast amount of researchers have reported a strong relationship between obesity and chronic kidney disease, and obesity can serve as an independent risk factor for kidney disease. The histological changes of kidneys in obesity-induced renal injury include glomerular or tubular hypertrophy, focal segmental glomerulosclerosis or bulbous sclerosis. Furthermore, inflammation, renal hemodynamic changes, insulin resistance and lipid metabolism disorders are all involved in the development and progression of obesity-induced nephropathy. However, there is no targeted treatment for obesity-related kidney disease. In this review, RAS inhibitors, SGLT2 inhibitors and melatonin would be presented to treat obesity-induced kidney injury. Furthermore, we concluded that melatonin can protect the kidney damage caused by obesity by inhibiting inflammation and oxidative stress, revealing its therapeutic potential.

## 1. Introduction

Obesity is prevalent in the context of the world, especially in developed countries where the obese population continues to increase [1]. Between 2005 and 2015, the obesity rate in the US was around 30–34%, and the prevalence of obesity in the UK was between 23–24% [2]. In general, the rates in low- and middle-income countries are relatively low compared to those in high-income countries. Probably due to the wide availability and accessibility of deeply processed foods, developing countries have shown an increase in the number of overweight and obese people; nevertheless, these processed foods are high in calories but lack nutritional value [3,4]. Obesity is now regarded as a growing and a disturbing global public health crisis, and China is the most affected country [5]. In particular, the adult overweight rate in China has almost tripled from 11.7 to 29.2% between 1991 and 2009 [6].

According to data from the World Health Organization (WHO), obesity is not only associated with an increased incidence rate, mortality and shortened life expectancy, but also the main risk factors for many chronic diseases, including type-2 diabetes, hypertension, cardiovascular diseases, dyslipidemia, non-alcoholic fatty liver disease and chronic kidney disease [7,8,9]. Actually, in addition to chronic kidney disease, there is also the classification of acute kidney injury (AKI) in kidney disease. There is evidence that patients with AKI have increased mortality or that AKI survivors are at further risk of developing CKD [10,11].

The rates of obesity have risen sharply in developing countries such as China and have had a major impact on the patterns of CKD [12]. On the flip side, a health diet is useful in the prevention and treatment of obesity and CKD [13]. Hence, in this review, the types and mechanisms of kidney damage caused by obesity are introduced in detail, along with the introduction of drugs for treating obesity-induced nephropathy. Moreover, melatonin is a better therapeutic drug.

## 2. Obesity-Induced Nephropathy

It has been nearly a century since the earliest research linking obesity and kidney disease [14]. Since then, the research on the relationship between obesity and kidney disease has carried forward. To reveal the relationship between CKD (estimated glomerular filtration rate <60 mL/min/1.73 m^2^) and obesity (defined by a body mass index (BMI) > 25 kg/m^2^), a large-scale study with 320,000 people found that the risk of end-stage renal disease (ESRD) increased in a stepwise fashion as BMI rose, after adjusting for the presence of hypertension, diabetes, a history of smoking and cardiovascular disease in patients [15]. A recent human experimental study showed that obese individuals had a higher risk of CKD compared to metabolically healthy normal-weight subjects, regardless of their metabolic status, even in the absence of remarkable metabolic abnormalities [16]. Moreover, obesity causes kidney disease in CKD and can also accelerate the loss of kidney function in patients with various primary renal diseases and decompensated renal function, such as IgA glomerulonephritis, kidney transplantation and diabetic nephropathy [17,18,19] (Figure 1). In short, accumulating evidence has suggested that obesity and kidney disease are closely related. Next, according to the histological level, kidney diseases caused by obesity will be divided into glomerular injury and renal tubular injury, and described further (Table 1).

### 2.1. Glomerular Injury

As the most familiar and well-known chronic renal complication of obesity, obesity-associated glomerulopathy (ORG) is a condition that has been identified as occurring specifically in obese individuals. Essentially, ORG is defined as a proteinuric renal disease in patients with BMI values of ≥30 kg/m^2^ and without clinical and histopathological evidence of other renal diseases [20]. Not surprisingly, in recent years, likely due to the increasing incidence of obesity and its complications in the general population, the prevalence of ORG is also increasing. On the one hand, the incidence of obesity-associated glomerulopathy increased more than tenfold from 1986 to 2015, and on the other hand, the average BMI of ORG patients was higher than that of normal people [22]. Especially, chronic kidney disease associated with diabetes was shown to be more common than glomerulonephritis, which had been the leading cause of ESRD in China [27]. Accordingly, it is more noticeable that individuals with obesity show the clinical and histopathological characteristics of typical ORG in early stage.

So how does the ORG specifically influence the kidney’s histology? The earliest study found that renal biopsy shows glomerular hypertrophy and focal segmental glomerulosclerosis (FSGS) lesions in patients with proteinuria caused by obesity [28]. Since then, a large number of studies on ORG have appeared, and it has been continuously verified that the previous studies are correct. Studies have shown that another characteristic histological change in patients with ORG may be the low glomerular density associated with glomerulomegaly [29]. In other words, a reduced total number of nephrons may lead to glomerular enlargement in patients with ORG. It has been hypothesized that glomerulomegaly enlargement may be due to an increased metabolic demand, and manifested functionally by an increased renal plasma flow (RPF) and glomerular filtration rate (GFR), which is generally considered to be a classical index for assessing the level of renal function [23,24]. A recent study found that the patients with ORG had a larger glomerular volume, a lower glomerular density and more glomerulosclerosis in the presence of similar total nephrons. Interestingly, the estimated total GFR was higher in ORG patients than in non-obese or obese controls [21]. Consequently, the results certified that glomeruli become enlarged with increasing obesity, and GFR also increased accordingly.

With a moderate to severe development of ORG proteinuria, glomerular hypertrophy commonly causes the lesions of FSGS [28]. FSGS in obesity-associated kidney disease is mediated by an adaptive increase in podocyte size that cannot keep up with glomerular expansion, which in turn leads to podocyte failure [22]. Specifically, in the early stage of ORG, obesity causes glomerular hypertrophy, which may cause glomerular podocytes to enlarge glomerular podocytes to cover the expanded glomerular capillary loops [30]. Afterwards, since glomerular podocytes cannot undergo division and dedifferentiation, a series of catastrophic events began, with foot process effacement, the detachment of podocytes and enhanced glomerular permeability. After the detachment of a large number of podocytes, the damage of FSGS followed. The glomerular clusters collapsed and were replaced by matrix deposition, accompanied by a total glomerular structural and functional loss during the final stage [31]. Nevertheless, not all patients with ORG have FSGS at the same time, which may be due to differences in obesity or renal insufficiency. For instance, Chen et al. observed no significant differences in glomerulus size and the number of cases with FSGS among groups with different BMI after the detection of 10,093 kidney biopsy samples from patients [32]. In short, under normal circumstances, ORG is morphologically diagnosed as glomerulomegaly with or without FSGS lesions (Figure 2).

### 2.2. Renal Tubular Injury

This could be due to the complicated structure of the renal tubules, which is not easy to analyze, and there are only a few studies on the structural and functional impairment of the renal tubules associated with obesity compared to the ORG. A study retrospectively found that the cross-sectional area of proximal tubular epithelial cells and the lumen of the proximal tubular cells were larger in proteinuric obese patients compared to non-obese patients with proteinuria [26]. Although tubular hypertrophy is a risk factor for the development of chronic kidney disease, the presence of tubular hypertrophy in obese individuals is meaningful because it maintains the structural basis for causing renal tubular hyperfunction. In a large number of animal experiments, obesity can cause renal tubular damage, including renal tubular hypertrophy, the appearance of lipid cytoplasmic inclusions, tubulo-interstitial inflammation and fibrosis [33,34] (Figure 3). In addition, studies have also shown that losing weight through drugs or daily diet adjustment can treat renal tubular damage in animal models [25]. For example, Qiao et al. found that melatonin can treat the edema/atrophy of renal tubules and proteinuria in obese rats [35].

In addition, as in diabetic nephropathy, the cause of tubular damage may be closely linked to obesity-related hyperfiltration. Specifically, the major site of renal glucose reabsorption is the proximal tubule (PT), and two active cross-linkers, sodium-dependent glucose transporter 1 (SGLT1) and sodium-dependent glucose transporter 2 (SGLT2), are expressed at the apical brush border of the PT [36]. Increased reabsorption of glucose and sodium by PT via SGLT1 and SGLT2 results in a reduced sodium load in the macula densa and distal tubule, which in turn activates tubulo-glomerular feedback, inducing increased preglomerular vasodilation and glomerular filtration rate [37]. On the whole kidney level and in normoglycemia, SGLT2 is responsible for the reabsorption of approximately 97% of the filtered glucose. Recently, SLGT2 inhibitors have been suggested to have a potentially important renoprotective effect, as evidenced by the fact that these drugs reduce the glomerular filtration rate and albuminuria in hyperfiltered diabetic patients [38]. The potential benefits of SGLT2 inhibitors in obesity-related kidney disease also raises an essential issue and is worthy of further investigation.

## 3. The Mechanism of Obesity-Induced Nephropathy

Adipose tissue affects the kidneys through a series of secretory factors, including cytokines, adipokines and metabolites, which are important to maintain a normal renal function. Yet, high levels of tumor necrosis factor-α (TNF-α), interleukin-6 (IL-6), adiponectin and angiotensin II and dysregulated metabolites are associated with obesity-induced kidney injury. These factors increase the changes of renal inflammation, insulin resistance and RAAS system activation, and eventually lead to renal injury (Figure 4).

### 3.1. The Inflammatory Response and Insulin Resistance

Adipose tissue is the primary place for the storage of excess energy in the form of triglycerides and is composed mainly of adipocytes [39]. As a result of changes in food intake or body metabolism, during energy balance, adipose tissue accumulates energy in lipid droplets rich in adipocytes. Generally, according to the structure and function of adipose cells, adipose tissue can be broadly divided into two types: white adipose tissue (WAT) and brown adipose tissue (BAT). WAT, the best-known adipose tissue in the organism, contains white adipocytes, which is where energy is stored [40]. A large number of studies have shown that the obese state is characterized by what has been called low-grade systemic inflammation, which is associated with WAT [41]. WAT occurs throughout the body in association with multiple organs, including the kidneys, and acts as an endocrine organ that secretes various biologically active substances that, because of their origin, are called adipokines or adipocytokines [42,43]. Under normal circumstances, the adipose tissue, accumulating around the kidneys, secretes adipokines, which are involved in immune response regulation and vascular homeostasis. However, in the obese state, adipose tissue might be releasing excess pro-inflammatory adipokines such as TNF-α and IL-6, and decrease beneficial adipokines, including leptin and adiponectin [44]. Among them, these pro-inflammatory cytokines are all involved in the process of cellular hypertrophy, extracellular matrix accumulation and renal fibrosis, and correlate positively with the urine albumin/creatinine ratio, which is a marker of renal injury [39,45]. These suggest that adipokines play an integral role in the development of obesity-induced kidney damage [46,47]—for example, elevated levels of TNF-α, and IL-6 are associated with a more accelerated progression of CKD [48,49]. Among these, the close associated of cytokines with obesity, TNF-α, IL-6 and adiponectin are further introduced.

TNF-α, as the first and most prominent inflammatory mediator in the inflammatory response process, induces a respiratory burst of neutrophils accompanied by the release of free radicals, while increasing the permeability of vascular endothelial cells, thus promoting renal injury [50]. In the same way, IL-6 is also a vital inflammatory factor in the initial stage of inflammation, and is highly expressed in various chronic inflammatory diseases. There have been many experimental studies and clinical observations showing that TNF-α and IL-6 also play a major role in the pathogenesis of chronic kidney disease through the induction of inflammatory processes. As a case in point, a study has shown that monocyte chemoattractant protein-1 (MCP-1) could stimulate the synthesis and release of TNF-α and IL-6 in renal cells, initiate programmed inflammation and further cause kidney damage [51]. TNF-α can induce the expression of MCP-1 in renal mesangial cells via the p38 mitogen-activated protein kinase (MAPK) signaling pathway [52]. On the other hand, the absence of TNF-α reduced obesity-related markers in the kidney to varying degrees, thereby decreased glomerular and tubular damage, and mitigated renal fibrosis [53,54]. Furthermore, as an integral part of the complex inflammatory network, TNF-α is able to initiate a cytokine cascade that controls the synthesis and expression of its linked cytokines and their receptors. The neutralization of TNF-α in rats with renal failure would lower the nuclear factor kappa B (NF-κB) activity and improve the release of nitric oxide (NO), which in turn has the potential to reduce renal inflammation and fibrosis, as well as oxidative stress [55,56]. Likewise, the blockade of the IL-6 receptor prevents the progression of proteinuria and renal lipid deposit as well as the mesangial cell proliferation associated with severe hyperlipo-proteinemia [57]. Thus, obesity may cause kidney damage by promoting the production of IL-6 and TNF-α.

On the flip side, there are also elements to adjust the activity of the inflammation to prevent extremes. Adiponectin is the most abundant adipocytokine in plasma, secreted predominantly by adipose tissue, and is thought to be involved in anti-inflammatory, anti-atherosclerotic and insulin-sensitizing effects in in vivo and in vitro experimental studies [58,59]. Unlike TNF-α and IL-6, adiponectin’s serum concentrations decrease with obesity [36], and recent studies have shown that a high serum adiponectin also signifies a renal dysfunction [60]. In addition, adiponectin can stimulate the expression of the anti-inflammatory cytokine IL-10 and reverse the pro-inflammatory effects of TNF-α, IL-6 and other cytokines in healthy persons and patients with type-2 diabetes and cardiovascular disease [61]. Adiponectin has now been identified to have a nephroprotective effect against lipotoxicity and oxidative stress by enhancing the AMPK/PPARα pathway and ceramidase activity. In fact, AMPK and PPARα are primary targets activated by AdipoR1 and AdipoR2, respectively [62]. In the regulation of biological metabolism, AMPK is a key molecule capable of mediating the intracellular signaling pathway of class O forkhead box (FOXO) proteins to regulate the expression of antioxidant enzymes [63] or to inhibit the NF-κB activation and anti-inflammatory effects [64]. The adipoR2-induced activation of PPARα enhances the oxidative capacity of the mitochondria to diminish oxidative stress and further facilitates the reduction of lipid accumulation in the target organ [65,66]. Therefore, the possible decrease in adiponectin levels by obesity can cause kidney damage in obese individuals, and the overproduction of adiponectin in adipose tissues can even exacerbate kidney inflammation in obese individuals [67].

There is, in addition, one further point to make. In the obese state, systemic chronic inflammation is strongly associated with insulin resistance. Several inflammatory biomarkers are elevated in the blood of obese patients with concomitant insulin resistance, such as C-reactive protein, IL-6 and TNF-α [68]. Similarly, experimental studies revealed that a dysfunctional lipid metabolism in adipose tissue results in an accumulation of circulating free fatty acids, initiating the inflammatory signaling cascades. As mentioned earlier, adipose tissue is an appreciable source of pro-inflammatory factors, which contribute to the development of insulin resistance and further adverse consequences [69,70]. The feedback effects of pro-inflammatory cytokines exacerbates this pathological state, promoting further cytokine secretion, and disrupts the insulin resistance in the same way [70]. In a study that evaluated insulin resistance by the insulin sensitivity index suggested that participants with CKD and obesity had a lower insulin sensitivity index [71]. In contrast, the adjustment for physical activity and diet partially attenuated the associations of CKD with insulin sensitivity and insulin [72]. The animal study also showed that modulating insulin signal transduction cascades could alleviate the high-fat diet (HFD)-induced insulin resistance and CKD in obese rats [73]. Consequently, in the development of obesity-related kidney disease, the hypothesis that insulin resistance plays a factor can be confirmed.

### 3.2. Renin–Angiotensin–Aldosterone System

The renin–angiotensin–aldosterone system (RAAS) is a kind of pressure-boosting regulation system produced by the kidney in the body, causing vascular smooth muscle contraction and water and sodium retention, and producing a pressure-boosting effect. Most components of RAAS have modest increases, predominantly in both animal and human visceral adipose tissue, which is the characteristic of obesity [74]. Tain et al. observed that the HFD diet was able to induce RAAS, that most components of RAAS did not differ with sex and that it caused renal damage in response to HFD exposure, although in another animal experiment a sex-dependent renal program within RAAS was found [75]. In the pathophysiology, one of the most influential elements in CKD and obesity is not circulating RAAS, but intrarenal RAAS, because of the role of intrarenal RAAS in sodium reabsorption and inflammation and fibrosis in the kidney [76]. Moreover, compared with non-obesity animals, the overactivation of RAAS activity in obese animals may be a factor causing glomerulomegaly and podocyte stress, eventually leading to glomerulosclerosis. Briefly, increasing evidence indicates that the activation of the RAAS plays a pivotal role during the progression of obesity-related kidney disease.

Concretely, renin near the ball of the juxtaglomerular apparatus granulosa cells releases a proteolytic enzyme and stimulates the plasma angiotensin I, which is further hydrolyzed into active angiotensin II [77]. Firstly, renin is the speed-limiting enzyme of the renin–angiotensin system, and its level can regulate the activity of RAAS. Secondly, angiotensin II, which is generated mostly in adipocytes and transported to the kidneys via the circulation, can promote the secretion of aldosterone. Angiotensin II exerts effects in the kidney through the activation of its receptors, including type-1 angiotensin II receptor (AT1) and type-2 angiotensin II receptor (AT2). Of interest, AT1 expression is increased in adipocytes and kidneys and vice versa [78], presumably owing to impaired macrophage polarization, increased macrophage infiltration and enhanced inflammation, and the deletion of the AT1a receptor (AT1AR, the major isoform of AT1) enhanced hyperlipidemia-induced structural kidney damage in mice [79]. However, the oral administration of a high dose of losartan (a selective AT1 receptor antagonist) slowed down the polarization of macrophages to pro-inflammatory macrophages and the infiltration of adipose tissue and kidney by inhibiting the up-regulation of AT1AR-mediated macrophages [80].

Besides, aldosterone is a steroid hormone (mineralocorticoid family) that enhances the kidney’s ability to reabsorb ions and water molecules. Aldosterone enhances the effects of angiotensin II, induces vascular inflammation and remodeling, and stimulates the mineralocorticoid receptors (MR) in the kidney [81]. MR activation may contribute to renal vasodilation and further impair the renal excretion function in obesity. When aldosterone activated MRs expressed on macula densa cells, the macula densa cells increased their production of NO and induced an increase in renal vasodilation and glomerular filtration [82]. On the contrary, blocking NO synthesis can eliminate aldosterone-induced hypertension [74]. Moreover, MR signaling can be stimulated by Rac1 and increase the levels of Rac1 in renal epithelial cells in the obese groups. From another perspective, the Rac1 inhibitor alleviates proteinuria and kidney damage [83]. High aldosterone levels can accelerate the onset and progression of renal diseases and predispose for diabetes and hyperlipidemia [84,85]. All in all, as described by the pathophysiological effects of MR, the inhibition of aldosterone with MR antagonists or by blocking the RAAS could slow the progression of renal disease.

### 3.3. Lipotoxicity

Because of lipid metabolism disorders, the amount of fat exceeds the normal range and is deposited on organs that should not be deposited, causing toxicity and damage to the organs, and this phenomenon is known as lipotoxicity [86]. At the whole-body level, lipotoxicity has been suggested to perform an important function in obesity, normally affecting the kidneys, liver, heart and skeletal muscle. Therefore, renal lipotoxicity may cause chronic kidney damage, a process that involves the accumulation of intracellular free fatty acids (FFA), triglycerides and toxic metabolites such as ceramides in renal glomerular and tubulointerstitial cells [87,88]. That said, the role of lipid dysregulation in inducing kidney injury in obese individuals has been widely reported [89]. More and more documents indicate that all cellular injury mechanisms of lipotoxicity can cause kidney injury, including inflammation, oxidative stress, fibrosis, alterations in intracellular signaling pathways and lipid-induced apoptosis [90]. Especially under obese conditions, in the presence of excessive reactive oxygen species (ROS) production or inflammatory response, the kidney will have a high chance of being exposed to cellular stress [91].

Nonetheless, previous studies have shown that the interaction between adipose tissue and the kidney, known as the adipose-kidney axis, acts in the response to injury and in the maintenance of kidney function [92]. Large amounts of lipid accumulation lead to lipotoxicity, where short-chain fatty acids rely on a passive diffusion of CD36, and fatty acids transport proteins to enter the cell [93], while long-chain fatty acids can only enter the cell through CD36 [94]. The CD36 pathway is involved in the uptake of free fatty acids and oxidized low-density lipoproteins, and its addition assists in mitigating lipotoxicity and proximal tubular and podocyte injury, while the ablation of CD36 attenuates renal injury [95]. Consistent with these findings, CD36 deletion almost completely reversed the ectopic renal lipid deposition in obese mice with inflammation and prevented renal injury [96]. Subsequently, FFA is internalized by carrier acylation, with a particular emphasis on the long-chain fatty acyl CoA formed by the esterification of the plasma membrane LCA-CoA synthase. LCA-CoA is transported through the carnitine shuttle to the mitochondrial matrix for fatty acid oxidation (FAO), and then, mitochondrial FAO produces mainly large amounts of ATP, which is the source of the increased ROS production that leads to early kidney damage [97,98]. Furthermore, there are several mechanisms: (1) in the obese state stimulation of malonyl CoA production by adipocytes due to insulin deficiency, which in turn promotes an increased oxidation of free fatty acids; (2) excessive production of saturated long-chain fatty acids; (3) activation of lipid-activated NADPH oxidase activity through the downstream production of ceramides, all of which can promote the accumulation of ROS [99]. As indicated in the previously, the effect of lipotoxicity on kidney may be due to the accumulation of ROS, which indirectly leads to kidney injury.

## 4. Treatment Drugs for Obesity-Induced Nephropathy

Drugs or treatments for other kidney diseases may not be appropriate for obesity-related kidney disease, given the specific and complex pathogenesis of obesity. To this end, dedicated studies are needed to test interventions designed to treat renal disease in patients with obesity. In previous studies of patients with obesity-related kidney disease, a series of pharmacological interventions have offered the possibility of slowing disease progression (Table 2)

### 4.1. RAS Inhibitors

How to suppress the renin-angiotensin system is a clinical issue that needs to be considered to slow down the progression of kidney disease. In general, RAS inhibitors are usually used to treat patients with proteinuria and diabetic nephropathy, as a way to reduce the incidence of proteinuria and CKD in ORG patients [69,111]. Moreover, compared to patients with normal BMI, obese patients may be more sensitive to the renoprotective function exerted by RAS inhibitors [112]. Representative drugs of RAS inhibitors include angiotensin-converting enzyme inhibitor (ACEI) and angiotensin-receptor blockers (ARB), which can inhibit and control high blood pressure, correct local hemodynamic abnormalities in the kidney, reduce proteinuria, reduce inflammation and protect the renal function [113] (Figure 5). Approximately 80% of the patients were taking ACEI or ARB at baseline as part of their standard care [114]. Particularly, in a post hoc analysis of renal disease treatment, it was shown that obesity predicted a higher incidence of nephropathy events, but the risk of nephropathy progression was significantly lower in obese patients on ramipril (a potent ACEI) than in non-obese patients [100]. An array of ARB has been invented to treat kidney disease in obese individuals as well. Telmisartan, an angiotensin II type-1 receptor antagonist, exerts a renoprotective effect by decreasing fat mass, promoting adipogenesis consumption and decreasing the perirenal WAT-derived leptin level [101]. Besides, the activation of MR can also ameliorate the adverse effects of obesity and metabolic syndrome [102], typically by adding spironolactone to an ACE inhibitor to reduce albuminuria and blood pressure in obese patients [115]. Nevertheless, it has also been suggested that, over time, the beneficial effects of RAS inhibitors may become depleted as some participants gain further weight [116]. Thus, this evidence leads to the speculation that RAS inhibitors are effective in inhibiting the development of obesity-induced kidney injury in the short term.

### 4.2. SGLT2 Inhibitors

Inhibitors of SGLT2 were first introduced as a new class of hypoglycemic agents, a new pharmacological option for the treatment of type-2 diabetes. As research continues to advance, SGLT2 inhibitors can also block the renal reabsorption of glucose and excrete the excess glucose from the urine, lower blood sugar and have protective effects on the kidney and the heart [36,103]. As a new target of hypoglycemic agent, SGLT2 also has the advantage of selective inhibition—a specific action in the kidney without significant effects on other tissues and organs [117]. At the same time, the benefits of SGLT2 inhibitors on the kidneys might not be explained solely by improving glucose control. As demonstrated, SGLT2 inhibitors may act on the kidneys by reducing fat and blood pressure [118], and are even able to reduce the SGLT2 expression in the high-glucose state [119]. They seem to have a great potential to improve the kidney damage caused by obesity. A large number of studies in patients and animals have reported that SGLT2 inhibitions were beneficial in regulating renal lipid metabolism, improving inflammation, preventing nephropathy and enhancing the cumulative effect of weight loss, natriuresis and osmotic diuresis [104,120,121]. Additionally, by using SGLT2 inhibition to reduce the reabsorption of glucose and sodium in the proximal tubule, glomerular hyperfiltration can be improved, and the progression of renal disease can be slowed [122]. The glomerular and tubular hemodynamic and hydrodynamic effects of SGLT2 inhibition play a major role in the afforded renoprotection. The inhibition of proximal tubular reabsorption abates glomerular hyperfiltration in obese non-diabetic subjects [123] (Figure 6). Thus, SGLT2 inhibition could theoretically have a renoprotective effect in obesity-induced glomerular hyperfiltration. In summary, a better understanding of the mechanism by which SGLT2 inhibitors exert their renal protective effects and whether this represents a unique effect has the potential to further improve the care of patients with obesity-related kidney disease.

### 4.3. Melatonin

Melatonin is an indole compound that is widely found in nature, from bacteria to humans [124]. In mammals, melatonin is one of the hormones produced by the pineal gland and, with its superior lipophilic effect, is readily released into the bloodstream and exerts its biological activity through melatonin receptors [105]. In terms of individual organisms, melatonin receptors are located throughout the body in multiple parenchymal organs, including the brain, retina, heart, blood vessels, liver, spleen, intestines and kidneys [125]. The wide distribution means that the application or lack of melatonin can affect many functions for human beings. As is well known, melatonin, a molecule that transmits photoperiod information to organisms, is predominantly produced at night and is essential in the regulation of normal circadian rhythms in the body [126]. It is necessary to emphasize that melatonin, in addition to its function in regulating the biological clock, plays a function in many organismal responses, such as anti-inflammatory, antioxidant and anti-apoptotic responses, the protection of the endothelial cell function and the inhibition of the sympathetic nerve’s activity [121]. When melatonin is added exogenously to the organism, the structure and function of the kidney are improved in models of kidney injury, including spontaneously hypertensive rats, rats with unilateral ureteral obstruction, rats with adriamycin-induced nephropathy and rats with 5/6 nephrectomy [106,107,127]. Of interest in this regard, the specific effect of melatonin on the kidney has recently been confirmed in an animal model with obesity [107]. More specifically, melatonin alleviates the inflammatory response after kidney injury and then further inhibits the subsequent activation of the oxidative stress pathways, since rats suffering from vascular hypertension with obesity showed an improvement in hypertension after melatonin treatment [108]. In the same way, melatonin exerts an antioxidant effect on the kidneys of obese rats with type-2 diabetes and Zucker diabetes, and its mechanism includes the increase of glutathione peroxidase (GPX) activities, the kidney superoxide dismutase (SOD) level and catalase (CAT) enzyme activity [35,109]. The above-mentioned SOD, CAT and GPX are used by cells to scavenge excess ROS, the production and release of which leads to inflammation, oxidative stress, tissue damage and disease progression [110]. Likewise, in obese kidney tissue of humans, oral melatonin therapy can significantly improve the antioxidative defense and lipid distribution. Therefore, the maintenance of a stable antioxidant enzyme activity and the exertion of its own antioxidant effect is an important reason why melatonin can be used as a therapeutic agent for obesity-related kidney diseases.

On the other hand, melatonin levels can also alleviate obese states. Evidence suggests that melatonin is able to regulate energy metabolism, including maintaining body weight, insulin sensitivity and glucose tolerance levels [128]. A large body of data shows that melatonin acts at every step of the energy balance, including food intake, energy storage and expenditure. In addition, the addition of exogenous melatonin after a pinealectomy may reduce body weight, reduce food intake and increase energy expenditure in brown adipose tissue [129]. Several mechanisms have been proposed for the physiological and pharmacological beneficial effects of melatonin in obesity, with particular attention to antioxidant and anti-inflammatory effects [130,131]. As previously mentioned, obesity is a chronic inflammation of the body, which can cause kidney damage through inflammatory pathways, and melatonin can reduce pro-inflammatory cytokines, adipokines, chemokines and adhesion molecules [130]. Some of these actions respond to transcription factor modulation, such as NF-κB, nuclear factor erythroid-2-related factor 2 (Nrf2) and others [132]. The latest study also found that melatonin’s protective actions included inhibition effects on inflammasomes [133]. Among them, the NLRP3 inflammasome is involved in obesity-related kidney disease development and may serve as a key modulator of obesity-related kidney disease [134]. The NLRP3 inflammasome can promote IL-1β (interleukin-1β) and IL-18 (interleukin-18) maturation [135]. Moreover, IL-18 can cause damage to the kidney by promoting the production of TNF-α, IL-1β and intercellular adhesion molecules via monocytes and macrophages [136]. Thus, the inhibition of NLRP3 inflammasome activation could attenuate obesity-related kidney disease.

In short, there are promising experimental data to suggest that melatonin, as an anti-inflammatory and immunoregulating molecule, a free radical scavenger, antioxidant and circadian rhythm regulator, could improve and even mitigate the progression of kidney disease in persons with obesity (Figure 7).

## 5. Perspective

At present, there are inadequate treatments for obesity-related kidney disease, and the main treatment for this disease is fat loss or the use of therapeutic drugs. For example, sleeve gastrectomy is used extensively in severely obese patients with renal failure [137]. Studies have shown that RAS inhibitors can delay the occurrence of microalbuminuria and its progression to macroalbuminuria in diabetic patients [138]. Startlingly, with the emergence of SGLT2 inhibitors, it can be used to improve kidney function [139]. Still, to evaluate the therapeutic effects of these drugs, a large number of experiments is required.

In view of previous reports, melatonin can be used as a drug to treat obesity-related kidney diseases [140]. It has been widely accepted that melatonin has a significant effect on regulating a series of biological functions—in other words, that it is a hormone that regulates circadian rhythms to relieve sleep disorders. From the results of previous animal experiments, it is known that taking melatonin can potently protect the kidney [108,141]. On the contrary, melatonin deficiency may increase the risk of developing numerous conditions, such as diabetes, obesity, cardiovascular disease and kidney disease. As an endogenous substance, melatonin works through the regulation of the endocrine system. It has its own metabolic pathway in the body and does not cause the accumulation of metabolites in the body. Moreover, its biological half-life is short, and so it has the characteristics of minimal toxicity [127]. Therefore, melatonin can be an adjuvant drug in the treatment of obesity-related kidney diseases. Simultaneously, there is a need for more precise melatonin treatments targeting the kidneys.

## 6. Conclusions

The obesity problem is becoming more and more serious all over the world, and its complications have also received more attention, especially obesity-related kidney disease. Kidney damages caused by obesity include glomerular hypertrophy and focal segmental glomerulosclerosis, renal tubular hypertrophy, renal tubular interstitial inflammation and fibrosis. Obesity promotes the secretion of pro-inflammatory adipokines by fat cells, which mediate inflammation and insulin resistance. Obesity also reduces the level of adiponectin, a fat factor involved in anti-inflammatory and insulin sensitization, thereby increasing kidney damage. In addition, obesity can also promote kidney damage through the progressive activation of RAAS and lipotoxicity. Clinically, both ACEI and SGLT2 inhibitors can treat obesity-related kidney disease, but both have their shortcomings. A large amount of data shows that melatonin has a potential renal protective effect on kidney damage, and it obviously has no major side effects. Melatonin plays a beneficial role in inhibiting oxidative stress and the inflammation of kidney cells through direct or indirect mechanisms. Therefore, the use of melatonin to prevent and treat obese or diabetic patients may be promising. However, further research is needed, such as the study of the dose-response relationship of melatonin, to expand the use of melatonin in clinical practice, prevention and treatment.

## Figures and Tables

**Figure 1 ijms-23-00747-f001:**
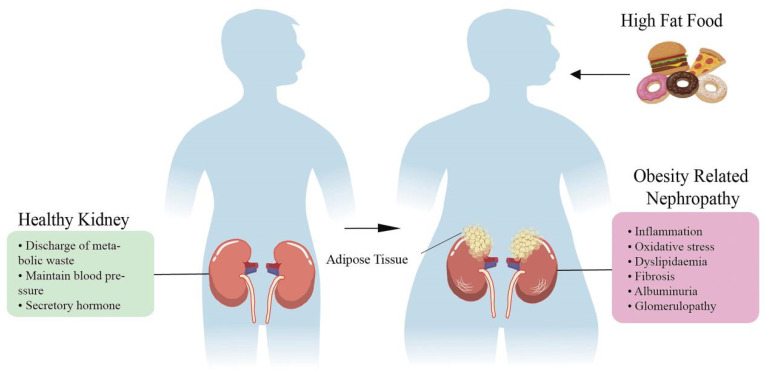
The influence of high-fat food on the progression of renal disease. The kidney has a urinary function. Essentially, dissolved materials in the liquid fraction of blood (plasma) are formed into a filtrate by the action of the kidneys. In this way, the kidneys remove waste and recycle filtered nutrients. In addition, the kidney can affect blood pressure by regulating the urine volume and blood volume or secreting vasoactive substances. The kidney also has the function of producing some hormones with endocrine activity, such as 1,25-dihydroxycholecalciferol [1,25-(OH) 2D3], renin, renal prostaglandin and erythropoietin. The body intake of high-calorie food increases the degree of fat accumulation. When a large amount of fat accumulates in the kidney, it accelerates a series of kidney damage, such as inflammation, oxidative stress and subsequent fibrosis. Long-term fat accumulation further leads to proteinuria, glomerular and tubular diseases.

**Figure 2 ijms-23-00747-f002:**
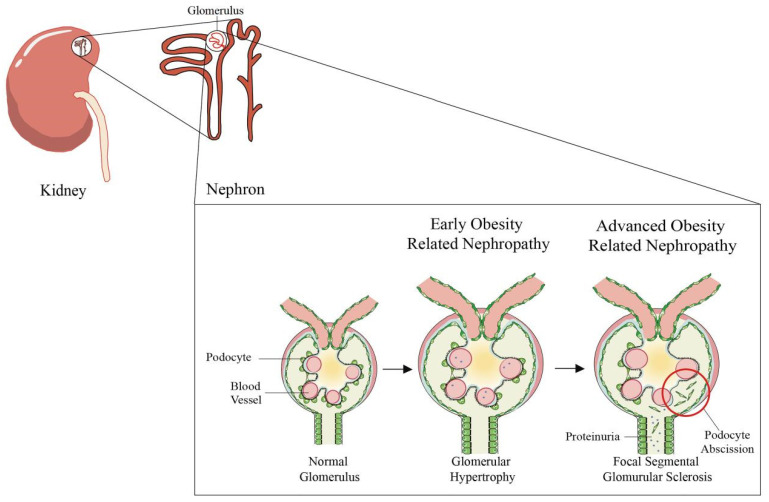
Pathophysiological changes of glomerulus in obesity-related nephropathy. Normal glomerular tissue consists of three parts: endothelial cells in the inner layer, glomerular basement membrane in the middle layer and podocytes in the outer layer. Glomerular hypertrophy and low glomerular density are associated with early obesity-related renal injury, followed by increased glomerular filtration rate and proteinuria. However, podocytes cannot catch up with the speed of glomerular expansion, podocyte failure, further regression and shedding, namely focal segmental glomurular selerosis (FSGS). Finally, glomerulosclerosis leads to severe renal damage.

**Figure 3 ijms-23-00747-f003:**
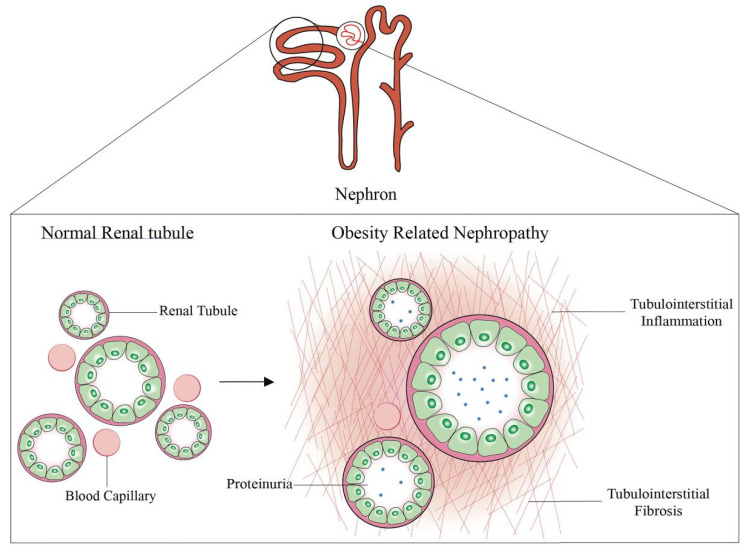
The changes of renal tubules in obesity-related nephropathy. Renal tubules are tubules surrounded by monolayer epithelial cells, which can reabsorb and secrete some components in urine. Obesity-related kidney disease is characterized by tubular hypertrophy, basement membrane thickening and then interstitial inflammation and fibrosis, and it may thus set the stage for renal failure later in life. In the right picture, red indicates renal interstitial inflammation, and the line indicates renal interstitial fibrosis.

**Figure 4 ijms-23-00747-f004:**
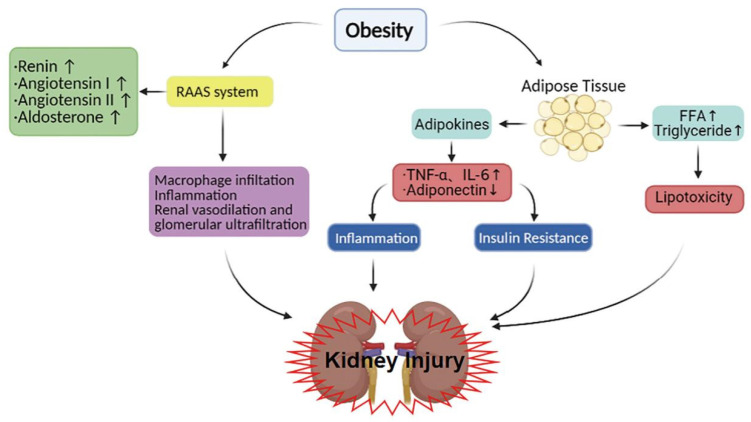
The mechanism by which obesity contributes to kidney injury. In obese individuals, adipose tissue secretes TNF-α, IL-6 and adiponectin. Among them, the pro-inflammatory factor TNF-α, the increase of IL-6 and the decrease of anti-inflammatory factor adiponectin promote the inflammatory response and insulin resistance, and cause the corresponding kidney injury. The lipids in the human body can be roughly divided into cholesterol and neutral fat (triglyceride, phospholipid). Free fatty acid is one of the substances decomposed from neutral fat. When the body is obese, the contents of triglyceride and FFA increase. Then, they are deposited in the kidney, causing toxicity and damage to the kidney and leading to serious lipid metabolism diseases. Similarly, RAAS can also be activated by obesity and promote macrophage infiltration, inflammation, renal vasodilation and glomerular ultrafiltration through a cascade reaction, leading to proteinuria and kidney injury. FFA, free fatty acids; IL-6, interleukin- 6; TNF-α, tumor necrosis factor-α. “↑” represents an increase, and “↓” represents a decrease.

**Figure 5 ijms-23-00747-f005:**
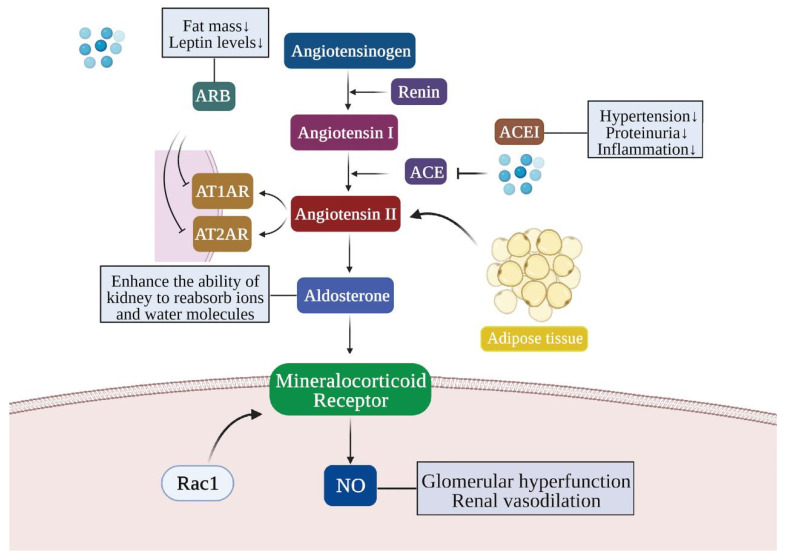
The role of the RAAS system and RAAS inhibitors in obesity-related kidney injury. The renin–angiotensin–aldosterone system (RAAS) is comprised of a series of peptide hormones and corresponding enzymes and is a significant humoral regulatory system. Renin secreted by renal juxtaglomerular cells can convert the angiotensinogen in the blood into angiotensin I without physiological activity. Angiotensin I forms angiotensin II under the action of an angiotensin-converting enzyme (ACE). Adipose tissue can promote the secretion of angiotensin II, which stimulates the secretion of aldosterone and promotes the reabsorption of water and sodium in renal tubules, improves the renal filtration rate and causes proteinuria. Aldosterone stimulates NO synthesis through the mineralocorticoid receptor (MR) or Rac1, which leads to glomerular hyperfunction, renal vasodilation and renal injury. RAAS inhibitors, ARB and ACEI correct the renal filtration efficiency by inhibiting the corresponding substance level in the RAAS system, and then alleviate renal injury. ACE, angiotensin converting enzyme; ACEI, angiotensin converting enzyme inhibitors; ARB, angiotensin receptor blocker; AT1AR, angiotensin type 1A receptor; AT2AR, angiotensin type 2A receptor; NO, nitrogen oxide; Rac1, ras-related C3 botulinum toxin substrate 1. “↓” represents a decrease.

**Figure 6 ijms-23-00747-f006:**
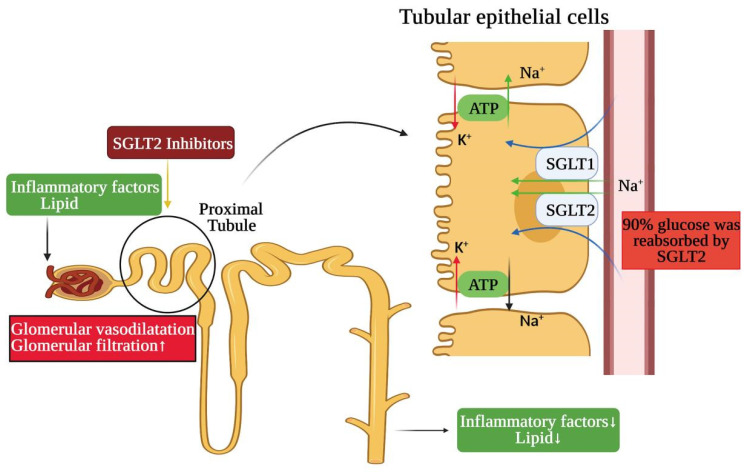
The mechanism of SGLTs and SGLT-2 inhibitors in obesity-related kidney injury. The reabsorption of glucose in the kidney mainly depends on sodium glucose cotransporters (SGLTs), through which glucose and Na + are transported across the membrane. SGLT-2 was mainly expressed in the kidney, while SGLT-1 was partially expressed in the kidney. About 90% of the glucose was reabsorbed by SGLT-2 in the S1 segment (the beginning of the proximal convoluted tubule and the first 1/3 of the convoluted part in the kidney) of the proximal convoluted tubule. However, SGLT-2 can also regulate inflammatory factors and lipids in proximal tubules, activate glomerular feedback and decrease the glomerular filtration rate, so as to slow down the progression of kidney disease. SGLT-1, sodium-dependent glucose transporters 1; SGLT-2, sodium-dependent glucose transporters 2. “↑” represents an increase, and “↓” represents a decrease.

**Figure 7 ijms-23-00747-f007:**
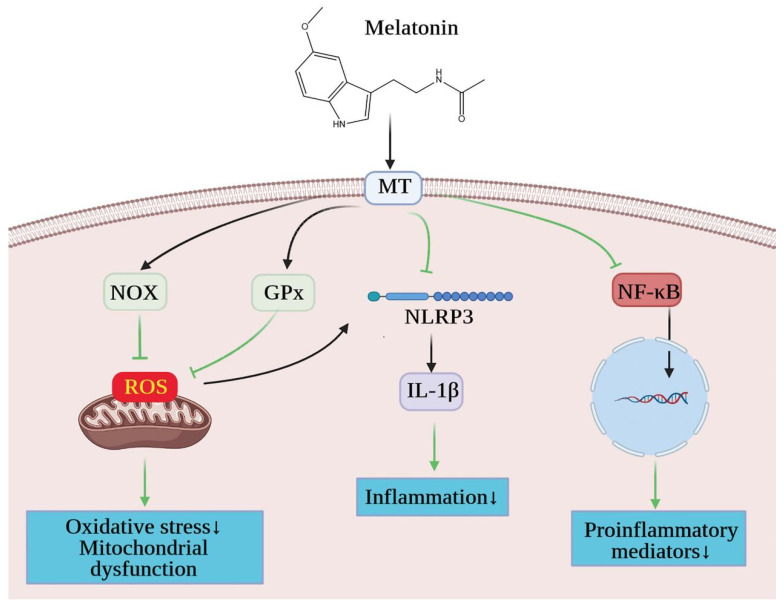
Therapeutic effect of melatonin on obesity-related kidney injury. Melatonin acts through melatonin receptors on the cells of the kidney. Melatonin can eliminate the excessive free radicals produced by mitochondria through antioxidant enzymes, alleviate mitochondrial dysfunction and play an antioxidant role. Melatonin can down-regulate NLRP3 inflammasome, which participates in the development of renal disease associated with obesity and may be a key regulator of obesity. Melatonin can also inhibit transcription factors including NF-κB, reduce the transcription of pro-inflammatory mediators, thereby reducing the level of inflammation, and fundamentally treat the kidney damage caused by obesity. GPx, Glutathione peroxidase; IL-1β, interleukin-1β; MR, melatonin receptor; NF-κB, nuclear factor kappa-B; NLRP3, NLR Family Pyrin Domain Containing 3; NOX, NADPH oxidase; ROS, reactive oxygen species. “↓” represents a decrease.

**Table 1 ijms-23-00747-t001:** Studies on obesity-related kidney damage.

Types of Kidney Injury	Pathological Damage in Kidney	References
Glomerular injury	Global or focal segmental glomerulosclerosis	[20,21]
	Glomerulomegaly (glomerular volume > 3.27 × 10^6^ μm^3^) with or without FSGS	[22,23,24]
	Low glomerular density	[23]
	Glomerulosclerosis	[21]
	Glomerular filtration barrier defect	[25]
Renal tubular injury	Tubular atrophy	[19]
	Proximal tubular epithelial hypertrophy	[26]
	Tubular cell apoptosis ↑	[25]
	Oxidative stress in tubular cells	
Others	The measured parenchymal volume and the estimated cortical volume of the kidneys ↑	[21]
	·Oxidative stress	
	·Enhanced renal lipid deposition	
	·Renal TGF-β1 and collagen type IV expression ↑	
	Glomerular and tubular lesion scores ↑	

‘↑’ represents an increase. FSGS, focal segmental glomerulosclerosis; TGF-β1, transforming growth factor-β1.

**Table 2 ijms-23-00747-t002:** Studies on the effects of different drugs on kidney injury and dysfunction.

Animal/Model	Drug/Dose	Renal Function	Kidney Injury	Reference
Human Obesity FSG	ACEI	Proteinuria ↓	Glomerulomegaly	[20]
Human Obesity RK or URA	ACEI	Proteinuria/renal insufficiency, serum creatinine ↓	N/A	[100]
Human Obesity ESRD	ACEI	Proteinuria ↓ BP burden →	N/A	[101]
Male Wistar rats Obesity	ACEI, telmisartan, (8 mg/kg/d, gastriclavage)	N/A	Local expression of adipogenesis markers and adiponectin ↑ IL-6, MCP-1 ↓	[102]
Human Obesity	Low-dose spironolactone + ACEI	Blood pressures, proteinuria ↓	N/A	[103]
Human Type 2 diabetes CKD	SGLT2 inhibitors, dapagliflozin (5 and 10 mg)	Blood pressures, proteinuria ↓ Serum creatinine ↑	N/A	[104]
Akita mice Diabetic	SGLT2 inhibitors, empagliflozin (300 mg/kg)	Systolic blood pressure ↓	GLUT1/GLUT2, inflammation, CD14,IL-6,TIMP2 ↓	[105]
Sprague-Dawleyrats 5/6 Nx CKD	Melatonin, (10 mg/100 mL, drink)	Blood pressure ↓	RAS, oxidative stress, Interstitial fibrosis ↓ Antioxidant activity ↑	[106]
Wistar Albino rat Unilateral ureteral obstruction (UUO)	Melatonin (1 mg/kg/day, intraperitoneal)	N/A	Oxidative stress, iNOS, p38-MAPK, NF-kB ↓ The development of leukocyte infiltration and interstitial fibrosis ↓	[107]
Boscat white rabbits Obesity	Melatonin, 1 mg/kg (sub-cutaneously)	Blood pressure, serum lipids, blood glucose, atherogenic index ↓	GSH-PX ↑	[108]
Wistar albino rats Renovascular hypertension	Melatonin (10 mg/kg/day)	Blood pressure ↓	Oxidative injury, plasma LDH, CK and ADMA levels ↓	[109]
Zücker rats Obesity Diabetic	Melatonin, (2 g/L and 20 mg/L, drink)	Proteinuria ↓	NOx, HFR, GSH/GSSG ↓	[110]
Wistar albino rats Diabetic nephropathy	MSCs + Melatonin (5 µM, in vitro)	N/A	Anti-inflammatory Anti-oxidation TNF-α, TGF-β1 ↓ IL-10, SOD, Beclin-1, glomerular sclerosis ↑	[46]

Abbreviation: ACEI, angiotensin-converting enzyme inhibitor; ADMA, asymmetric dimethylarginine; BP, blood pressure; CK, creatine kinase; CKD, chronic kidney disease; Drp1, dynamin-related-protein 1; ESRD, end stage renal disease; FSG, focal segmental glomerulosclerosis; GLUT, glucose transporter; GSH, glutathione; GSH-PX, glutathione peroxidase; GSSG, L-glutathione oxidized; HFR, hydroxyl free radicals; IL-6, interleukin-6; IL-10, interleukin-10; iNOS, inducible nitric oxide synthase; LDH, lactate dehydrogenase; MAPK, mitogen-activated protein kinase; MCP-1, monocyte chemoattractant protein-1; Mfn2, mitofusin2; MSC, mesenchymal stem cell; NF-kB, nuclear factor kappa-B; NOx, NADPH oxidase; Nx, nephrectomized; N/A, not applicable; Opa1, optic atrophy 1; RAS, renin-angiotensin system; RK, remnant kidney; SGLT2, sodium-dependent glucose transporters 2; SOD, superoxide dismutase; TBARS, thiobarbituric acid-reacting substances; TGF-β1, transforming growth factor-β1; TIMP2, tissue inhibitor of metalloproteinases-1; TNF-α, tumor necrosis factor-α; URA, unilateral renal agenesis; UUO, unilateral ureteral obstruction. “↑” represents an increase, “↓” represents a decrease, and “→” represents no effect.

## Data Availability

Not applicable.

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
