# Peer review of "Kidney Damage Caused by Obesity and Its Feasible Treatment Drugs"

_ijms, 2022, doi:10.3390/ijms23020747_

Round 1
Reviewer 1 Report
The review is interesting since the topic is current. The authors extensively reviewed the existing literature and provided the data in the manuscript accompanied by high-quality pictures.
I have the following comments:
- Although generally written correctly, the manuscript should be re-checked by the authors since there are some linguistic and type errors. There is no need for external evaluation since the authors can easily accomplish this task.
- In my opinion, the possible role of melatonin, compared to RAS and SGLT2 inhibitors, is overemphasized.
- Regarding glucotoxicity and FFA toxicity, a more detailed description of the molecular mechanisms, which eventually lead to increased ROS production by the mitochondria, would be welcome. I suggest evaluating some reviews written by Brownlee M. for DM angiopathy (Giacco F et al., Circ Res 2010; 107: 1058–10).
- Another mechanism that may explain part of the renoprotective effects of SGLT2i is the reduced glucotoxicity in renal tubular epithelial cells (Eleftheriadis et al., Int Urol Nephrol 2020; DOI: 10.1007/s11255-020-02481-3).
- At some points, too much information is provided. I suggest organizing the text so that the meaning of studies showing similar results to be summarized in one sentence, referencing all the related studies in the end, instead of using one sentence per study.
- In such an extensive review manuscript, some mistakes inevitably occur. For instance, NADPH oxidase is not protective against oxidative stress, as written, but a main source of ROS. Please, re-check.
Author Response
Answer to the Reviewers’ Comments
Dear reviewers of International journal of mechanical sciences:
We are very glad to receive your mail. Thank you for your letter and for your comments on our manuscript entitled "Renal damage due to obesity and its possible therapeutic agents". Those comments are valuable and very helpful for revising and improving our paper. Based on your comment and request, we have made extensive modification on the original manuscript. Detailed modifications are attached below.
Finally, we acknowledge the reviewer’s comments and suggestions very much, which are valuable in improving the quality of our manuscript.
Corresponding authors:
Name: Yulan Dong
E-mail: ylbcdong@cau.edu.cn
Point 1: Although generally written correctly, the manuscript should be re-checked by the authors since there are some linguistic and type errors. There is no need for external evaluation since the authors can easily accomplish this task.
Response 1: We gratefully appreciate for your valuable suggestion. The manuscript has been rechecked and the linguistic and type errors have been carefully corrected.
Pleases see the full text.
Point 2: In my opinion, the possible role of melatonin, compared to RAS and SGLT2 inhibitors, is overemphasized.
Response 2: We totally understand the reviewer`s concern. Since the potential therapeutic role of melatonin is becoming widely recognized with the discovery of studies, in this section we emphasize melatonin compared to RAS and SGLT2 inhibitors.
Pleases see "4.3 Melatonin" section in the manuscript.
Point 3: Regarding glucotoxicity and FFA toxicity, a more detailed description of the molecular mechanisms, which eventually lead to increased ROS production by the mitochondria, would be welcome. I suggest evaluating some reviews written by Brownlee M. for DM angiopathy (Giacco F et al., Circ Res 2010; 107: 1058–10).
Response 3: We gratefully thanks for the precious time the reviewer spent making constructive remarks. We describe in detail the molecular mechanisms that ultimately lead to more ROS production by mitochondria in the manuscript.
Pleases see lines 361-366 on pages 9-10 in the manuscript.
Point 4: Another mechanism that may explain part of the renoprotective effects of SGLT2i is the reduced glucotoxicity in renal tubular epithelial cells (Eleftheriadis et al., Int Urol Nephrol 2020; DOI: 10.1007/s11255-020-02481-3).
Response 4: Thank you for your rigorous nice suggestion. We learned that part of the mechanism of the nephroprotective effect of SGLT2i is to reduce glucose toxicity in renal tubular epithelial cells, and added it in the manuscript.
Pleases see lines 50-51 on page 14 in the manuscript.
Point 5: At some points, too much information is provided. I suggest organizing the text so that the meaning of studies showing similar results to be summarized in one sentence, referencing all the related studies in the end, instead of using one sentence per study.
Response 5: We feel sorry for the inconvenience brought to the reviewer. Loose phrases throughout the text have been carefully revised and condensed. If further grammatical changes are needed, we will choose to perform language editing of the MDPI.
Pleases see the full text.
Point 6: In such an extensive review manuscript, some mistakes inevitably occur. For instance, NADPH oxidase is not protective against oxidative stress, as written, but a main source of ROS. Please, re-check.
Response 6: Thank you so much for your careful check. The role of NADPH oxidase mentioned in the manuscript has been re-examined and corrected.
Pleases see lines 364-366 on page 10 in the manuscript.
Reviewer 2 Report
Dear authors
The review focus on the effect of obesity in kidney damage and highlight the possible role of RAS inhibitors, SGLT2 inhibitors and melatonin. The review is well written, scientifically interesting to the reader and describes the role of inflammation and oxidative stress as mediators of kiedney damage. Compared to the other recents review papers the novelty is the role of SLGT2 inhibitors and the possible role of melatonin
However concerning the adiponectin pathway I suggest to
- Improve the paragraph 3.1 with recents references on the role of adiponectin in both kidney disease and obesity. In this regard I suggest these 2 recent works: PMID: 33255520 ; PMID: 32221363
- In figure legend 2 correct the space as the others
Author Response
Answer to the Reviewers’ Comments
Dear reviewers of International journal of mechanical sciences:
We are very glad to receive your mail. Thank you for your letter and for your comments on our manuscript entitled "Renal damage due to obesity and its possible therapeutic agents". Those comments are valuable and very helpful for revising and improving our paper. Based on your comment and request, we have made extensive modification on the original manuscript. Detailed modifications are attached below.
Finally, we acknowledge the reviewer’s comments and suggestions very much, which are valuable in improving the quality of our manuscript.
Corresponding authors:
Name: Yulan Dong
E-mail: ylbcdong@cau.edu.cn
Point:The review focus on the effect of obesity in kidney damage and highlight the possible role of RAS inhibitors, SGLT2 inhibitors and melatonin. The review is well written, scientifically interesting to the reader and describes the role of inflammation and oxidative stress as mediators of kidney damage. Compared to the other recent review papers the novelty is the role of SLGT2 inhibitors and the possible role of melatonin
However concerning the adiponectin pathway I suggest to
Improve the paragraph 3.1 with recent references on the role of adiponectin in both kidney disease and obesity. In this regard I suggest these 2 recent works: PMID: 33255520 ; PMID: 32221363
In figure legend 2 correct the space as the others
Response:We gratefully thanks for the precious time the reviewer spent making constructive remarks. We learned about the role of adiponectin in kidney disease and obesity by studying two recent papers, and adding to the latest manuscript. Firstly, We learned that high serum adiponectin is a biomarker of renal insufficiency from PMID: 32221363 and added in the manuscript.
Pleases see lines 247-248 on page 7 in the manuscript.
Secondly, we learned from PMID: 32221363 that adiponectin can inhibit the inflammatory response via the AdipoR1-pAMPK-NFκB pathway and added in the manuscript.
Pleases see line 257 on page 7 in the manuscript.
Round 2
Reviewer 1 Report
The authors addressed all issues.